# A Novel Efficient Dynamic Throttling Strategy for Blockchain-Based Intrusion Detection Systems in 6G-Enabled VSNs

**DOI:** 10.3390/s23188006

**Published:** 2023-09-21

**Authors:** Lampis Alevizos, Vinh Thong Ta, Max Hashem Eiza

**Affiliations:** 1School of Psychology and Computer Science, University of Central Lancashire (UCLan), Preston PR1 2HE, UK; 2Department of Computer Science, Edge Hill University, Ormskirk L39 4QP, UK; tav@edgehill.ac.uk; 3School of Computer Science and Mathematics, Liverpool John Moores University (LJMU), Liverpool L3 3AF, UK; m.hashemeiza@ljmu.ac.uk

**Keywords:** vehicular social networks, 6G technology, blockchain, intrusion detection

## Abstract

Vehicular Social Networks (VSNs) have emerged as a new social interaction paradigm, where vehicles can form social networks on the roads to improve the convenience/safety of passengers. VSNs are part of Vehicle to Everything (V2X) services, which is one of the industrial verticals in the coming sixth generation (6G) networks. The lower latency, higher connection density, and near-100% coverage envisaged in 6G will enable more efficient implementation of VSNs applications. The purpose of this study is to address the problem of lateral movements of attackers who could compromise one device in a VSN, given the large number of connected devices and services in VSNs and attack other devices and vehicles. This challenge is addressed via our proposed Blockchain-based Collaborative Distributed Intrusion Detection (BCDID) system with a novel Dynamic Throttling Strategy (DTS) to detect and prevent attackers’ lateral movements in VSNs. Our experiments showed how the proposed DTS improve the effectiveness of the BCDID system in terms of detection capabilities and handling queries three times faster than the default strategy with 350k queries tested. We concluded that our DTS strategy can increase transaction processing capacity in the BCDID system and improve its performance while maintaining the integrity of data on-chain.

## 1. Introduction

Vehicular Social Networks (VSNs) represent a unique form of localised mobile social networks that exploit vehicular communication links and offer travellers the opportunity to engage in social activities along the road. Direct inquiry from others with a similar experience in proximity over social networks tends to be the most convenient and efficient approach to acquiring up-to-date, specialised, and domain-specific information for travellers [1]. Moreover, a TripAdvisor survey showed that 76% of travellers share their travel experience including photos and clips via social networks and 52% do that while on the road [2]. Based on the physical and social distances of users, many applications have been proposed in the context of VSNs such as UberPool [3] and Verse [4]. VSNs can be formed using different approaches such as the use of infrastructure, through an Internet connection, or in an ad hoc manner via vehicle-to-vehicle communication [5].

By the year 2030, the official roll out of the sixth generation (6G) networks is expected to meet the demands of mobile communications, when the current 5G networks will have reached their limit [6]. The 6G networks are envisaged to provide global coverage and Tbps-level transmission data rates for applications such as Virtual Reality (VR), 3D videos, and Augmented Reality (AR). Besides higher data rates, 6G should provide lower latency, higher connection density, and near-100% global coverage in comparison to 5G. This has the potential to revolutionise VSNs by providing enhanced connectivity, ultra-low latency, and enormous data capacity. The ultra-low latency communications would enable real-time interactions between vehicles in a VSN, which can facilitate various safety applications such as collision avoidance systems, cooperative adaptive cruise control, and emergency notifications. In addition, the high data transfer speeds can support high-resolution video streaming, AR, and VR applications in vehicles. With 6G, passengers in VSNs can have an enhanced experience in social interactions and gaming.

Unfortunately, besides these benefits, the fact that vehicles and their embedded devices in a VSN are connected through many social applications would increase the risk of lateral movement by attackers who can start compromising a single vehicle or device (i.e., the weakest link) in a VSN as a starting point and then extend the attack to other vehicles and devices. Furthermore, if there is an opportunity for different VSNs to connect to each other, the extent of lateral movement can be even greater. This is the case in “corporate” VSNs, for instance, when public transport buses, operated by a bus company, form VSNs on the road and the passengers can use social apps installed in the tablet attached to the back of their seats. Passengers can interact and play with fellow passengers on the same bus or other buses in the same VSN. Drivers and vehicles can interact with others and the stations to obtain real-time information about the routes. If the attacker manages to compromise and control a social app, the attacker can extend the attack to the other apps in the device and other devices via lateral movement. The effect of lateral movement can be more devastating in the case of ad hoc VSNs, where a compromised vehicle can leave one VSN and join another on the road.

In the face of this problem, Distributed Intrusion Detection Systems (IDSs) have been proposed for the Internet of Vehicles (IoV), Vehicular Ad hoc Networks (VANETs), and Internet of Things (IoT) to cope with their dynamic nature. In these approaches, instead of centralised detection servers or cloud, the vehicles and the roadside units function as intrusion detection sensors (e.g., through the standalone IDSs installed in them). This enables the concept of collaborative distributed intrusion detection (CID) when anomalies can be detected based on the logs and network data shared by the vehicles and roadside units. For example, in the case of Sybil attacks, vehicles monitor the network and share information about observed vehicle identities, communication patterns, and message content. If a vehicle detects multiple identities associated with a single vehicle, it flags the vehicle as suspicious, and this information is disseminated to other vehicles. When multiple vehicles report the same vehicle as a potential Sybil node, a consensus is reached, and appropriate countermeasures can be taken (e.g., isolating/ignoring the Sybil node). However, despite the benefits of collaborative distributed intrusion detection, it only works well if all vehicles are honest, unselfish, and benign. Selfish vehicles can refuse to share information with others, while dishonest and malicious vehicles or roadside units (e.g., controlled by the attacker via malware infection) can intentionally disseminate incorrect or incomplete information.

### 1.1. Main Scope and Contributions

To rectify the problem above, we propose a Blockchain-based Collaborative Distributed Intrusion Detection (BCDID) system in the context of VSNs, which provides data integrity and immutability for the shared information among vehicles and roadside units. The scope of our study assumes a corporate VSN where vehicles in the same company form the network. These companies can be, for example, taxi, travel agency, or bus companies. Each VSN maintains a permissioned or corporate blockchain that contains a whitelist of the cryptographic hashes of benign social applications running in the devices attached to the back of the passengers’ seats as well as in front of the drivers. Whenever a new social application is installed or changes have been made to an existing one, the system calculates the cryptographic hash of the new or modified application. The newly calculated hash will be committed to the blockchain for whitelisting. Note that this blockchain-based intrusion detection concept is based on the approach proposed in our previous work [7] that detects and prevents malicious applications on endpoints. The main contributions of this paper are as follows: We propose a novel ledger-query strategy, named “Dynamic Throttling Strategy”, that not only works best for the BCDID use case but can be leveraged widely by blockchain networks when simple key–value queries with substantial amounts of data and users are the basic characteristics of these networks.We show that the proposed BCDID can be applied in the new VSN context with some modifications to deal with the large number of nodes since the approach proposed in [7] was only tested in a small-scale network. Hence, we address the performance limitations in large-scale application contexts like VSNs and conduct performance analysis to show how our proposed solution can cope and excel in the case of a large number of endpoints in VSNs.

### 1.2. Literature Review—Collaborative Intrusion Detection Systems

Deploying Intrusion Detection Systems (IDSs) is a well-known approach to effectively detect intrusions based on the anomaly caused by malicious or compromised devices. Hence, it is one of the most promising solutions for the problem in discussion. However, implementing a standalone IDS is often insufficient in the case of large networks due to the substantial number of false positives and negatives. Shortcomings of standalone IDS systems have been studied by Fung et al. [8], Duma et al. [9], and Weizhi et al. [10]. As a result, DCIDSs have been proposed to improve the efficiency and availability of standalone IDSs. Collaborative Intrusion Detection Systems (CIDSs) overcome the limitations of standalone IDSs mentioned in [11], where IDS nodes can be installed in the network but do not share information with each other, and therefore their detection capability is more limited compared to that of CIDSs. On the other hand, CIDSs are usually a network of cooperative IDSs that leverage collective knowledge to achieve improved accuracy in detecting intrusions. Furthermore, Distributed CIDS (DCIDSs) are implemented to deal with attacks such as Distributed Denial of Service (DDoS) attacks which traditional IDSs cannot tackle effectively. Wu et al. [12] showed that in practice, compared to a standalone IDS setting, CIDSs can reduce the number of missed alarms to one from seven cases, and the number of false alarms were also reduced in their test system.

For these reasons, DCIDSs can be seen as one promising approach to detecting anomalies in IoVs, VANETs, and IoT. The authors in [13] presented a CIDS designed specifically for VANETs. The proposed system employs a trust-based approach and utilises vehicle-to-vehicle communication to detect and mitigate various attacks in VANETs, such as Sybil attacks and black hole attacks. Experimental results demonstrated the effectiveness of the CIDS in improving intrusion detection accuracy. Zhou et al. [14] proposed a DCIDS based on invariants to identify betrayal attacks in VANETs. Their approach is a reputation-based cooperative communication method and a so-called cluster head vehicle selection method based on the global reputation state, traffic density, and link life. Zhang et al. [15] presented a machine learning-based privacy-preserving DCIDS for VANETs. To detect intrusion in VANETs, the authors introduced an approach based on the alternating direction method of multipliers (ADMM) to a class of empirical risk minimisation and proposed a method of dual variable perturbation to provide dynamic differential privacy.

In [16], the authors proposed a data-driven IDS for IoV by analysing the link load behaviours of the Roadside Unit (RSU) against various attacks causing fluctuations in traffic flows. The detection approach is based on a Convolutional Neural Network (CNN) with the features such as link loads, and it detects the intrusion aimed at RSUs. Anzel et al. [17] proposed a multilayer perceptron (MLP) neural network to detect intruders or attackers on an IoV network. In addition to these, many other AI-based intrusion detection methods have been proposed in the literature as discussed by Man et al. [18].

Intrusion detection approaches have also been widely proposed for Internet of Things applications (e.g., [19,20,21]). Alshahrani [22] proposed a collaborative intruder detection system that detects malicious activities in IoT devices. The framework gathers information from four main layers, namely, the IoT layer, network layer, fog layer, and cloud layer to monitor and analyse the network traffic among IoT devices. The authors in [23,24] proposed blockchain-based solutions to detect malicious vehicles and IoT devices. The main difference between our work and the work proposed in [23] is that in the latter case, the malicious behaviour of the vehicles is detected using machine learning (neural networks), while in our case we focus on the lateral movement among the endpoints installed inside the vehicles. The authors in [24] addressed the problem of indoor navigation and proposed a new secure communication approach based on blockchain, which is different from the objective of our paper.

#### 1.2.1. CIDSs Architectures

While CIDSs and DCIDS can reduce the rate of false negatives and positives because of the network data shared among the IDS nodes, they also have some limitations such as (1) the increased attack surface as now the attacker may target more IDS nodes, and effectively protecting all the nodes is a challenging task; (2) the data shared among IDS nodes may be inconsistent or incomplete due to lack of trust among the nodes, as well as selfish or compromised nodes. 

In general, CIDSs can follow a centralised, hierarchical, or peer-to-peer architecture. In the first case, decision-making is made by a central server which collects and processes data sent from all IDS sensor nodes. Intrusion detection is based on algorithms that use aggregated information and correlated events. In hierarchical architectures, IDS sensor nodes are organised in multiple tiers, and local data processing and analysis are completed in each tier. The analysis results of each tier are forwarded to the higher-level tiers until they reach the top tier. The main advantages of this architecture include better scalability and load distribution. Compared to the centralised case, depending on the number of tiers, we may expect some delay with intrusion detection. Finally, in the case of peer-to-peer architecture, IDS sensor nodes share data directly with each other to collectively make decisions. This architecture facilitates decentralised intrusion detection based on correlated alerts and avoids the single-point-of-failure problem in the centralised case.

#### 1.2.2. Alert Correlation

In the hierarchical and peer-to-peer architectures, alert propagation and correlation follow different approaches. For example, Garcia et al. [25] proposed a hierarchical CIDS architecture that correlates alerts from IDS nodes using secure multicast. In their approach, local IDS, called “prevention cells”, detect and record attacks locally, which are then exchanged between the local IDSs. In addition, Dash et al. [26] proposed a collaborative host-based IDS approach which detects network intrusion using distributed probabilistic inference. Dain et al. [27] proposed a scenario-based probabilistic approach using three variations of Bayesian networks. In this method, the detected events and anomalies are categorised into attack scenarios. Cuppens et al. [28] introduced Language to Model a Database for Detection of Attacks, an attack description language aiming to correlate alerts based on the specification of the pre-and post-condition of a target system. Cheung et al. [29] proposed Correlated Attack Modelling Language, which allows lower-level specification of attack scenarios and anomalies. Templeton and Levitt [30] proposed an attack specification language JIGSAW for DCIDSs, which specifies attacks on the threat event-type level rather than attack scenarios.

#### 1.2.3. Alert Trustworthiness

Besides the effectiveness of sharing data with each other, the quality and completeness of the exchanged information are crucial. Solutions to tackle this problem have been proposed in the literature. For example, to detect message tampering and forging, the authors in [31] proposed a digital signature and cryptographic hash-based authentication solution for alert messages in a peer-to-peer CIDS architecture. In addition, to detect selfish IDS nodes sending incomplete/incorrect information, Chen et al. [32] proposed the use of a “Web of Trust” between participating nodes, in which the quality of the exchanged information can be measured by the reputation of the nodes.

Blockchain-enabled IDSs can be used to ensure the integrity and trustworthiness of alert messages, and several works can be found in the literature as discussed in the review paper [33]; however, these are mostly theoretical concepts without extensive real-life experiments and/or implementation. For example, the authors in [34] proposed a CIDS concept based on a permissioned blockchain concept, where a set of alert messages are defined as transactions within the blockchain. Li et al. [20] presented the so-called Collaborative Blockchained Signature-Based Intrusion Detection System (CBSigIDS) for trust management. The authors in [35] proposed a Collaborative IoT Anomaly Detection (CIoTA) framework. To make the concept suitable for IoT devices, the authors designed a lightweight approach and demonstrated it using a testbed of 48 Raspberry Pi’s.

## 2. Blockchain-Based Collaborative Distributed Intrusion Detection (BCDID) System

In this section, we present our blockchain’s building blocks and elements and discuss the steps and intrusion detection mechanism. Again, note that the system presented in this section is based on our previous work in [7], but we show how the BCDID system improves upon our previous work, and it can be applied in the new VSN context. The main novelty and contribution of this work is the new proposed mechanism to improve the effectiveness of the blockchain-based IDS in case of many endpoints, which is relevant in the VSN context. The proposed BCDID system uses a private and permissioned blockchain architecture where each company builds and maintains its own private blockchain to detect intrusion or anomalies in its VSN. This approach preserves the privacy and confidentiality of corporate data and controls the access of the participating nodes, users, and administrators. For evaluation in this study, we selected the Hyperledger Fabric (HLF) blockchain platform.

### 2.1. Building Blocks and System Model

Each corporation has a fleet of vehicles that can be buses, cars, trains, or even planes. We assume that the endpoints are the devices mounted into the seats of the passengers for entertainment during their journeys. The endpoints for passengers have a set of social and game applications installed, which are regularly updated, and applications are also added or removed to meet the budget of the corporation but at the same time provide up-to-date entertainment opportunities for the passengers. The applications can run on Windows, Linux, Android, or iOS platforms, depending on the specific device the corporation installs. We present the architecture of the proposed BCDID system below, assuming each corporation has a headquarters, different divisions or branches, and a fleet of vehicles with endpoints installed in them.

**Organisations and Peers:** For simplicity, we explain our system with a corporation with a headquarters and one branch, for which we define two organisational elements in the blockchain architecture, OrgHQ and OrgBR. This concept can be extended straightforwardly to a corporation with multiple branches and multiple corporations. OrgHQ represents the headquarters and OrgBR represents a single branch of it. Each organisational OrgHQ and OrgBR runs its own peer OrgHQ Peer (which is owned and operated by the headquarters) and OrgBR Peer (operated by the branch), respectively. Both peers host their own ledger alongside their Smart Contracts, also known as Chaincode. Their ledger immutably records all transactions generated by smart contracts.

**Ledger:** In our proposed concept, the ledger stores the current hashes of the endpoints mounted in the vehicles. The ledger also stores the past hashes as a history of transactions that eventually resulted in the current values, providing a reliable source of the chain of events in case of a required software update on an endpoint. The ledger comprises two separate segments, namely, the world state database and the blockchain. On one hand, the world state database contains the current values of the hashes produced from the endpoint. On the other hand, the blockchain records all changes leading up to and including the current value of the world state database, in the form of transactions. Afterwards, transactions are “placed” inside blocks and ultimately appended on the blockchain which enables better understanding of historical changes that led to the current value in the world state database. Blocks enclose ordered transactions and are bounded cryptographically with the previous and next block, ultimately forming a chain of transaction logs in the form of chained blocks of transactions.

**Orderer**: The Orderer is a special node responsible for ordering transactions, creating a new block of ordered transactions, and distributing the newly created block to all peers on the communication channel, thus always keeping ledgers on “OrgHQ Peer” and “OrgBR Peer” consistent.

**Channel:** Within the corporate blockchain network, channels are communication mediums for OrgHQ and OrgBR (and their components). For the two peers of each organisation to respectively join the channel, an identity is required. For every transaction that is executed via the channel, the peers and entities must first acquire authentication and gain authorisation.

**Consensus:** Our proposed BCDID architecture is based on Hyperledger Fabric, thus inherently relying on a deterministic consensus algorithm. Determinism in the context of blockchains means that if one enacts the same steps in a pre-defined order, the same results as anybody else who follows the exact process should be achieved. This eventually provides a guarantee that any block validated by peers is correct and final. The consensus mechanism in our case can be divided into the following three phases: (A) endorsement, (B) ordering, (C) validation and commitment.

**Client:** The client is the actual application (or even a set of applications) that interacts with the blockchain network.

**Certificate Authorities:** CAs are responsible for managing user certificates such as user registration, user enrolment, and user revocation. X.509 standard certificates can be used. The network setup is based on a permissioned blockchain network; therefore, only permitted users can:query peer ledgers and access information, orinvoke, namely, create new transactions.

**Transaction proposal:** An administrator or user proposes a transaction to submit a new executable’s hash for whitelisting through the “Client” which is signed by the user’s or administrator’s certificate. Next, the proposal is sent to the pre-defined endorsing peers “OrgHQ Peer” and “OrgBR Peer” through the channel. Endorsing peers perform a sequence of verification checks such as whether the proposed transaction has not been submitted in the past, the validity of digital signatures, as well conformance with the communication channel writer’s policy.

**Generalising to multiple organisations:** In this case, more peers can join the channel, and the single orderer would most likely get overburdened with tasks such as distributing blocks of transactions. However, a secondary orderer can always be added or even a cluster of orderer nodes ideally. Regarding the possible network congestion due to block distribution overhead, the concept of leading peers is utilised as a mitigating measure. For this concept to be triggered an organisation (e.g., “OrgHQ”) would need more than one peer, and as such, for example, one peer would take the leading role while the other would function as an endorsing peer.

#### Operation Processes

The goal of the BCDID system is to effectively detect, and prevent where possible, attacks on the endpoints. To achieve this, the main processes are defined as follows (see Figure 1):**Process 1—New endpoint enrolment:** At the beginning and whenever the installation of new media devices in the vehicles happens, these new endpoints are enrolled.**Process 2—Import new apps on-chain:** This process utilises the so-called “CreateAsset” app function and chain code, which enables newly hashed social application information to be transferred and recorded on-chain, providing immutability.**Process 3**—**Verify existing apps on-chain:** This is the core element of whitelisting the “benign” applications, where the presence of an app’s information on the chain is verified, and the execution of the app can be denied if the information is not on the chain. For this, we define the so-called “AssetExists” app function.**Process 4—Query for a specific app(s):** To manually verify the on-chain presence of applications or request certain information for incident triaging, we define “GetAllAssets” or “ReadAsset” apps functions and chaincodes. These allow an administrator to query the ledger for specific information.**Process 5—Update existing app(s) information:** To update certain information fields of applications on-chain, we define the app function and chaincode called “UpdateAsset”.**Process 6—Detection and prevention triggers:** If an app is trying to execute without the relevant data being present on-chain, then an alert is generated. We generate two types of alerts: (1) when an app is trying to execute without the relevant data being present on-chain, and (2) when an admin-owned app (see Process 7 below) is trying to execute. Both cases indicate a potential intrusion. Nonetheless, alerts and rules can be configured and further refined at a later stage to include countless cases.**Process 7—Transfer app(s) ownership:** To transfer the ownership of apps on-chain, we define the app function and chaincode called “TransferAsset”, which creates a sequence and reference in the form of transactions.

### 2.2. Blockchain-Based Collaborative Distributed Intrusion Detection Mechanism

To detect and block attackers from lateral movement as early as possible, the proposed intrusion detection mechanism is based on hash-based Blockchain-enabled whitelisting. Each endpoint (tablet, media device, or computer) has a set of social applications installed in them, and the newly installed or approved versions of the applications are hashed beforehand. For security reasons, SHA-256 or SHA-512 can be used for file hashing purposes. Depending on the specific operating system platform these apps run on, the apps can have the “.apk” extension (in the case of Android) and “.ipa” (iOS), and in the case of Windows, they can be “.exe”, “.bin”, “.msi”, etc.

The precision of our proposed approach is based on the detection accuracy, namely, how likely compromised apps/malware are not detected but are allowed to run in an endpoint. To examine this, we built a testbed and considered 31% of the attack techniques in the MITRE based on the ATT&CK adversary tactics (https://attack.mitre.org/) to launch file-based and fileless attacks against an endpoint. We showed that for the file-based attacks, unlike a machine learning IDS-based approach, our BCDID method achieved 100% correctness for both detection and prevention for each attack. However, for the fileless attacks, the accuracy rate is around 63% as 17 out of the 46 examined attacks were able to carry out the injection of apps/tools directly into the memory, bypassing detection.

### 2.3. Performance Issues

In this section, we discuss the main problems and challenges related to performance when implementing the proposed BCDID in practice. Then, in Section 2.4 we discuss the related works that address Hyperledger Fabric performance problems, followed by our proposed novel method to overcome this performance issue in Section 2.5 and Section 2.6. Several essential functions take place within the BCDID ecosystem. Processes 1 and 2 do not have a time constraint attached to them. The blockchain network administrator(s) per organisation can build the necessary application whitelist before allowing access to the corporate resources in advance. The BCDID prototype can onboard 200 users within approximately 75 min, generating 1 million successful transactions in total, with a rate of 220 Transactions Per Second (TPS). Therefore, user onboarding, firstly, is usually not a time-bounded task, and secondly, even if an organisation has hard deadlines on user onboarding, with an extremely limited resourced prototype like ours, it could onboard 1300 new endpoints per working day (assuming 8 h equal a working day).

Thus, our first performance evaluation workload generation and measurement are focused on Process 3. Process 3 is where the decision-making on whether an application is allowed to be executed or not transpires. Consequently, this is also a key point for Process 6, whereas if an application is not allowed to execute, a potential intrusion detection alert needs to be raised. On the contrary, if the outcome of Process 3 is positive, namely, the query returns the required value, then the application will be allowed execution. This is likely the first potential performance bottleneck, as hundreds or even thousands of users are anticipated to execute applications simultaneously, thus translating into hundreds or thousands of transactions on the backend system. Before diving into system bottleneck analysis, it is imperative to understand the two BCDID application-peer interactions, namely ledger-update versus ledger-query transactions. Table 1 summarises the individual consensus-related actions while showing where the invoke or query is required.

A ledger query transaction is far more lightweight than ledger-update (invoke) since it does not need to engage multiple peers nor the ordering service. Therefore, it is best suited for low-latency read-only activities, without the necessity to record data on-chain. So, to answer the question “*what happens when hundreds or thousands of users try to execute an application and thereby start a ledger-query transaction simultaneously*”, we need to breakdown the exact steps of a ledger-query transaction. In the case of a ledger-query transaction, the transaction proposal and endorsement consist of three discrete steps. These are part of the client application and peer interaction. Specifically, in our BCDID ecosystem, the client application represents the user. Thus, the sequence for an endpoint with a user having a valid identity is as follows:**Transaction proposal:** A user belonging to OrgHQ executes a single application chrome.exe, which automatically triggers the “AssetExists” chaincode and therefore submits a signed response with the user’s certificate-transaction proposal to the endorsing organisation OrgHQ peer(s).**Transaction execution:** peer0 belonging to OrgHQ executes the chaincode “ReadAsset” specified in the proposal and generates a proposal response which contains the read-write set. The response is signed by peer0 and is sent back to the user.In case the output matches the input, namely, the current hash of chrome.exe is identical to the one existing on-chain, chrome.exe will be allowed execution.In case the output of “ReadAsset” returns a hash mismatch, chrome.exe will be denied execution.Additionally, an intrusion alert will be triggered, and Process 6 will begin (see Figure 1).Transaction endorsement: the transaction will be executed repeatedly for each organisation required by the chaincode endorsement policy. Responses are collected and signed.

We measured the performance of the above-mentioned ledger-query Step 2, assuming a group of 100 up to 1000 users attempt a simultaneous execution of the Chrome web browser. Chrome requires 350 different executables to be queried prior to allowing execution, which we measured on the user endpoint. Our observations are shown in Figure 2a,b.

The BCDID’s CPU and memory resources are quickly depleted as transactions (Tx) increase per user group. Notably, for the first 300 users, the resources seem to be enough; however, when we add 100 more users (400 in total), the TPS and the resources overall reach their limit. From that point onwards, TPS decreases while the time to complete significantly increases. The BCDID’s CPU and memory resources are quickly depleted as transactions (Tx) increases per user group. 

**Observation 1:** a performance bottleneck occurs when 400 or more users attempt simultaneous execution, which hinders user experience by significantly increasing the launch time of an application, and thereby the waiting time.**Observation 2:** even before the 400-user threshold, the CPU already operates at 90% usage on average, while the more load we add, the faster it reaches 100% of usage. This causes a resource utilisation problem that ultimately adds up to Observation 1.

Although the BCDID provides a great intrusion detection and prevention ratio against APTs, as demonstrated in [7], its performance is of utmost importance as it is directly connected with the user experience. Namely, the more time a ledger-query transaction takes to complete, the more equal the amount of time a user will have to wait for the requested application to execute. Thus, not only does this hinder user experience, but it also potentially affects business operations as well. Therefore, it is imperative to improve the performance of the BCDID ledger-query transaction, achieving the optimal peer specifications usage while minimising the time to respond. 

### 2.4. Hyperledger Fabric Performance Related Work

To understand the related work and existing solutions to the performance problem, we review the work of other scholars on the subject. The first version of Hyperledger Fabric v0.6 achieved less than 1k TPS [36,37] due to its core components architecture. In continuation, significant performance improvements and changes in core architectural components were introduced that achieved far better TPS. The membership service provider (MSP) caching was one of them. The MSP allows for deserialised certificate storage to reduce the overhead for crypto operations [38]. A second one is the parallel validation system chaincode (VSCC) which reduces the time for crypto operations by validating block signatures in parallel [39]. The TPS was improved even further by eliminating the lock contentions to access the cache, an improvement related to MSP caching, and thus TPS increased up to 2.5k [40]. Androulaki et al. [41] used SSDs for databases and block-file storages and achieved 4k TPS using Hyperledger Fabric v1.0. Gorenflo et al. [42] introduced four main architecture optimisations in Hyperledger Fabric v1.4, namely, separating data from metadata, parallelism and caching transaction data, memory hierarchy exploitation for faster data access, and resource separation for peers, to eventually achieve 20k TPS. Sousa et al. [43] designed, implemented, and evaluated a Byzantine Fault Tolerance (BFT) ordering service, ultimately reaching up to 10k TPS while writing time on-chain was measured to half a second with peers being distributed across continents.

Innovation through optimisation, rearchitecting of components, the combination of software and hardware configurations, and other methodologies have been studied extensively in the category of ledger-update transactions. The same does not apply to the ledger-query transactions, however. Although there are several studies on the subject, they focus on or around the same improvements but with different approaches. For example, Gupta et al. [44] presented two models with variations to create temporal indexes on the fabric. 

Yongqiang Lu et al. [45] proposed two different index building methods. These methods called temporal index based on state databases (TISD) and temporal index based on files (TIF). Both works seem promising; however, there are two drawbacks specific to our use case. Firstly, their experiments used a small number of entities (Yongqiang Lu et al. [45] being the largest one used 520, specifically), yet the maintenance and production of the mentioned indexes proved to be a rather complex methodology. In our case, we assume at least 50 million entities. Thus, the production and maintenance of indexes throughout state, history, and index databases would require significant effort to always keep up-to-date. Moreover, indexing approaches would introduce a security gap in our BCDID, namely, a potential breach of the index would compromise the entire notion of the BCDID integrity. Other relevant studies have performed measurements on the validation phase with either GolevelDB or CouchDB and a combination of the two (as being native choices of HLF), and even some have proposed the introduction of an entirely different database other than the two natively available in Hyperledger Fabric and moving the querying function off-chain [39,46,47]. Such approaches might offer some improvements on the query response; however, they would defeat two of the core BCDID’s notions, namely, remove trust from the endpoint and place it on-chain. Additionally, the performance of GolevelDB versus that of CouchDB, when it comes to simple key–value pair queries, has been extensively studied, and GolevelDB offers the best performance. In the case of BCDID, we use simple key–value pairs, where complex queries are not the case as well; thus, other databases would only increase complexity and cost without significant performance benefits [48].

### 2.5. A Novel Dynamic Throttling Approach to Enhance the BCDID’s Performance

The relevant literature and our observations provide a clear research direction. We first analyse how Hyperledger Fabric assigns peers for transaction execution. Second, we propose a novel ledger-query strategy named “Dynamic Throttling Strategy”, which not only works best for the BCDID use case but can also be leveraged widely when simple key–value queries with substantial amounts of data and users are the basic characteristics of a blockchain network. 

Peer selection is governed by HLF’s query strategies. The SDK provides 2 native strategies to evaluate transactions. Once defined through “DefaultQueryHandlerStrategies”, it is used for all transaction evaluations. If no strategy is defined, the default option of “PREFER_MSPID_SCOPE_SINGLE” is applied. The 2 native strategies with a variation in the fall-back method for each are described below:**PREFER_MSPID_SCOPE_SINGLE:** evaluates all transactions using the first peer of an organisation that can provide a response. It only switches to another peer if the first peer fails to provide a response for any reason. If the organisation has no peers, then it falls back to all peers specified in the network configuration file.○**MSPID_SCOPE_SINGLE:** follows similar principles as per the above strategy; however, in case of no available peers or no peers at all, the fall-back strategy is to fail exit rather than falling back to all peers within the network configuration file. **PREFER_MSPID_SCOPE_ROUND_ROBIN:** evaluates a transaction based on the list of peers, starting with the first on that list. Peers will be engaged in order until a response is received or until all peers have been engaged. On the next query, the second peer on the list will be engaged first, and then continue in the list of peers until a response is received. This is an incremental loading strategy that distributes the workload among all responding peers.○**MSPID_SCOPE_ROUND_ROBIN:** follows similar principles as per the above strategy; however, it will fail exit when there are no peers available on the organisation’s list, rather than falling back to all peers within the network configuration file.

### 2.6. Dynamic Throttling Strategy (DTS)

To overcome the difficulties with the existing strategies and based on Observations 1 and 2 in Section 2.3, we propose a novel dynamic throttling strategy. The strategy is based on two pillars: **(1) the peer environment indexing and monitoring and (2) an algorithm.**

**The peer environment indexing and monitoring** is as shown in Figure 3. We define three peer status tags based on our previous observations and measurements of 100–1000 users and up to 350k Tx’s. The peer status definition allows for a generalisation at this point based on the observed loading pattern of a single peer. Nonetheless, a 10% safety threshold for peers tagged as “available” is added. This means that peers in the mentioned state will still be able to manage queries without failures, as a single request will never consume more than 10% of a single peer resource. We also introduce a separate VM that hosts the index of peers’ reports of their CPU and RAM consumption in real-time to the peer index. Peers report in real-time their CPU and RAM consumption; therefore, the index controls the query distribution based on the algorithm. The response is sent directly back to the user.**The dynamic throttling algorithm**, as shown in Figure 4 above, is embedded in the blockchain network operating as our own query strategy. The users perform a substantial number of queries in parallel using the “D_THROTTLE” strategy, which triggers the dynamic throttling algorithm. Upon the successful identification of the first available node in a ready state, the index will assign the query to a subject node, while the node id will be registered, and the index will be updated (update +). Once the query is executed, results are returned directly to the user, and the node sends a cooldown signal updating the index (update −) with the current resource status. In the case that a node in a ready state is not available, the same flow will occur, but the index will search for the first available node this time. Conversely, if there is no node in an available state, the index returns error code −1, and the auto scale-up procedure begins to add resources to nodes currently marked as overloaded and updates the index accordingly.

## 3. Results and Discussion

We set up a test environment using the following machines and parameters detailed in Table 2. To begin with, both variations of the two core strategies are automatically descoped since within a private permissioned blockchain-based ecosystem, the parties (organisations) do not inherently trust each other. Equally, the peers of another organisation are not to be trusted and queried unless explicitly stated through an endorsement policy. In Section 2.3. Performance Problem Statement, we evaluated the performance of the BCDID based on the first and default strategy “PREFER_MSPID_SCOPE_SINGLE”. The results show that a single peer strategy is not suitable for the BCDID use case.

The last available native strategy is “PREFER_MSPID_SCOPE_ROUND_ROBIN”. Round Robin is a static algorithm that works in a circular and ordered manner. Each peer is assigned a query without any form of prioritisation. Furthermore, assuming 100 users query peer0 and peer1 of Org1 through the chaincode to evaluate Chrome’s hash presence on-chain (transaction), the algorithm distributes the load equally to both peers. In the meantime, we assume that a third peer is added on OrgHQ (peer3—OrgHQ) and another 50 users try to query the ledger against another application (e.g., outlook.exe). In this case, since the Round Robin algorithm works in a cyclic manner, we will have peer1 and peer2 managing the initial 100 requests, while peer3 will manage 50 requests. Hence, Round Robin fails to distribute the query load in an efficient routine. This is visualised in Figure 5.

As a result, we will always have the capacity to execute queries; however, this will be without unnecessarily overspending computing or money resources. Our strategy prioritises nodes in ready state first, progressively loading the cluster of nodes which eventually solves the problem identified during our first workload performance test. To verify this claim, we conducted the same initial experiment with the same parameters (viz. same number of users and applications in use), However, we utilised our “D_THROTTLE” algorithm and query strategy this time and we observed the following:**Observation 1:** by adding more nodes and using the “D_THROTTLE” algorithm, we managed to increase considerably the number of Transactions Per Second (TPS) up to 1991, which is double that of the default strategy; see Figure 6b.**Observation 2:** CPU and memory performance on all peers showed a declining trendline for all four peers. Moreover, none of the peers exceeded the 80% threshold to be marked as overloaded, while the average CPU usage for all peers ranged between 40% to 46%. This demonstrates a significant improvement in resource handling compared to the default strategy; see Figure 6a.**Observation 3:** the overall time to completion comparison chart highlights (1) that the dynamic throttling strategy is significantly faster than without it, and (2) that the more transactions received, a much smoother increase in time is anticipated, compared to the default query strategy; see Figure 7b.**Observation 4:** the time to completion per additional 50k queries is a steady line ranging between 17 to 18 s while using dynamic throttling, proving effective and efficient load balancing. While using the default strategy, however, the time to completion for the first 100 users measured up to 50 s, and it is evident that the peer is quickly allocating resources to complete the transactions but while reaching its threshold the time increases drastically beyond 60 s. Furthermore, once the peer finalises several transactions and frees some resources, there is a slight improvement in performance, yet again allocating all resources and quickly reaching the threshold, eventually leading to delays, as the pattern suggests; see Figure 7a.

## 4. Conclusions

In this paper, we proposed a novel efficient Dynamic Throttling Strategy (DTS) to improve the effectiveness and performance of Blockchain-based Collaborative Distributed Intrusion Detection (BCDID) systems in 6G-enabled VSNs. The developed BCDID can detect lateral movements and prevent attackers from compromising other devices if they were successful on a single device in the network. To show the performance problem of BCDID, we conducted an experiment to identify and set the baseline metrics to show how the existing ledger query strategies are not suitable. Therefore, our DTS was proposed to overcome this problem and implemented in the “D THROTTLE” algorithm. The evaluation results showed that the transaction processing capacity significantly increased, with a maximum of 1991 Transactions Per Second (TPS) achieved. CPU and memory performance on all peers showed a declining trend line, indicating improved resource handling. The overall time-to-completion comparison chart demonstrated that the DTS was significantly faster than the default query strategy. As more transactions were received, the increase in processing time was much smoother with the DTS. When comparing the time to completion per additional 50k queries, the DTS showed a steady line ranging between 17s and 18s, indicating effective and efficient load balancing.

Considering the contribution of the DTS to BCDID performance, there are several potential future directions for further research and development. First, enhancing the scalability of the system is crucial, particularly as the number of transactions and users increases. Additionally, investigating the integration of advanced machine learning algorithms for anomaly detection and threat intelligence can enhance the system’s ability to detect and prevent sophisticated attacks. Furthermore, exploring interoperability with other security systems and protocols can facilitate seamless integration and information sharing between different security components. Finally, conducting real-world deployment and testing of the BCDID in diverse organisational settings would provide valuable insights into its practical effectiveness and potential areas for further improvement.

## Figures and Tables

**Figure 1 sensors-23-08006-f001:**
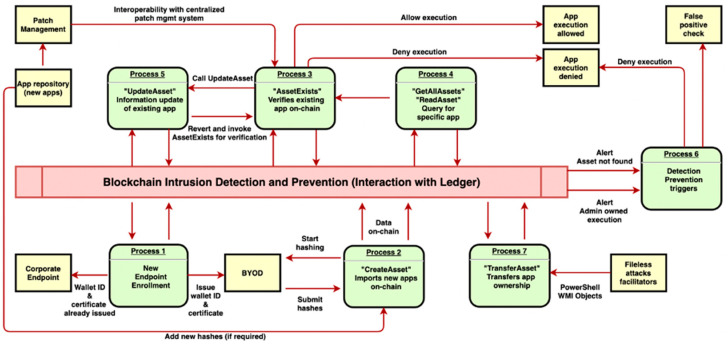
Operation processes blueprint.

**Figure 2 sensors-23-08006-f002:**
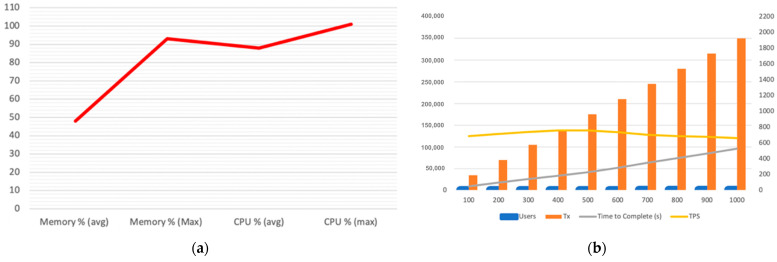
(**a**) CPU and memory performance; (**b**) Time to complete and TPS per user group.

**Figure 3 sensors-23-08006-f003:**
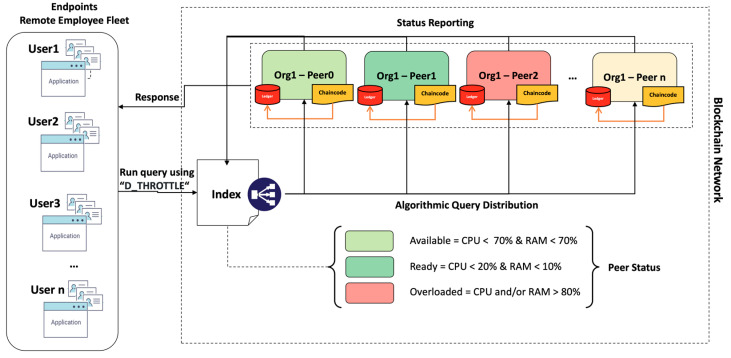
Peer environment indexing and monitoring.

**Figure 4 sensors-23-08006-f004:**
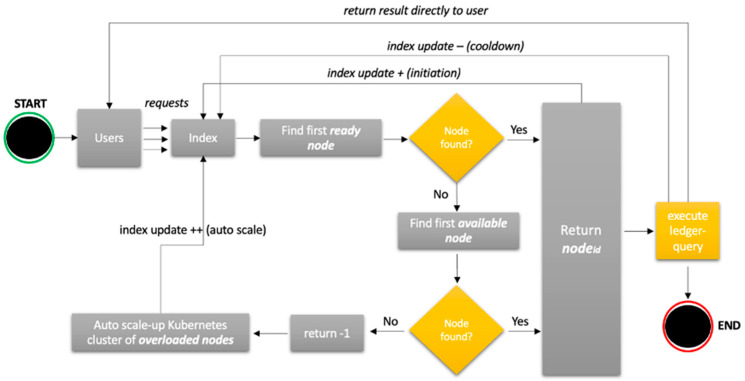
Dynamic throttling algorithm flowchart.

**Figure 5 sensors-23-08006-f005:**
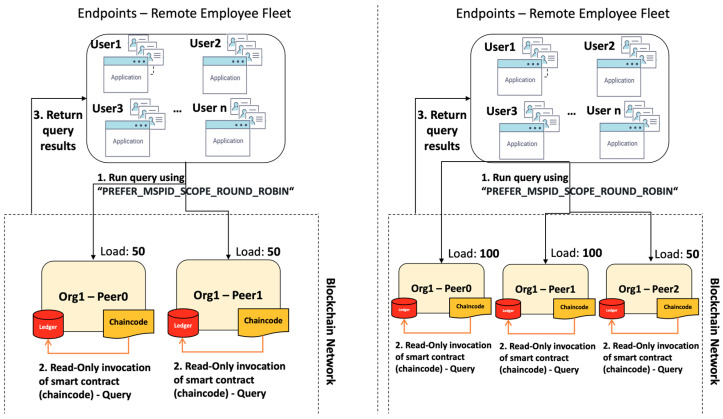
PREFER_MSPID_SCOPE_ROUND_ROBIN drawback.

**Figure 6 sensors-23-08006-f006:**
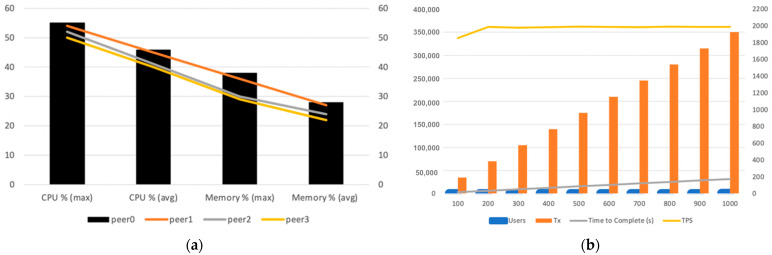
(**a**) CPU and memory performance using D_THROTTLE besides the percentage between 0 and 60; (**b**) Time to complete and TPS per user group using D_THROTTLE. We considered the user numbers from 100 till 1000, the TPS from 0 to 2200, and the transaction numbers from 0 to 400 K.

**Figure 7 sensors-23-08006-f007:**
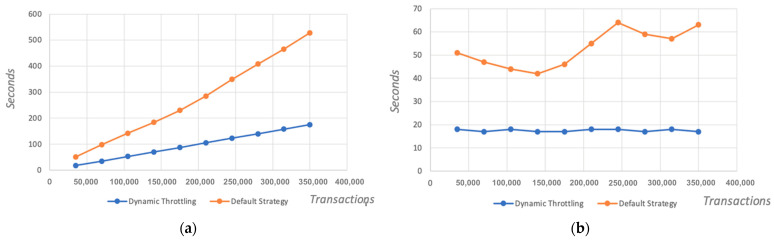
(**a**) Overall time to completion—seconds vs. transactions split; (**b**) Time to completion per transaction group—seconds vs. transactions split. The horizontal lines show the number of transactions we ran in each case, while the vertical lines show the times in seconds.

**Table 1 sensors-23-08006-t001:** Invoke versus Query.

Action vs. Transaction Method	Invoke	Query
Results in the update of world-state DB	Yes	No
Transaction data saved on-chain	Yes	No
Requires responses from multiple peers	Yes	No
Triggers ordering service and block creation	Yes	No

**Table 2 sensors-23-08006-t002:** Blockchain Lab Specifications.

Operating System	Ubuntu 20.04.3 LTS (GNU/Linux 5.11.0-27-generic x86_64)
Hard Disk Drives	25 GB
Central Processing Unit	2.22 GHz Quad Core Intel Core i7-4770HQ
Random Access Memory	6 GB
Software	Git, cURL, Docker, JQ, GO, Hyperledger Fabric 2.3, Ubuntu 20x basic installation with advanced package tool (APT) and APT essentials

## Data Availability

The data for this research is available upon request.

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
