# Peer review of "A Novel Efficient Dynamic Throttling Strategy for Blockchain-Based Intrusion Detection Systems in 6G-Enabled VSNs"

_sensors, 2023, doi:10.3390/s23188006_

Round 1
Reviewer 1 Report
In this paper, the authors proposed a blockchain-based collaborative distributed intrusion detection system with a dynamic throttling strategy to detect and prevent attackers’ lateral movement among devices in vehicular social networks.
The paper’s scope is within the scope of the journal, and it presents an original contribution. The abstract is somehow sufficient to give useful information about the paper’s topic. The proposed technique is described. The paper is somehow well structured and written, and the text is clear and easy to read. The conclusions are consistent with the evidence and arguments that are presented. However, there are some comments we recommend the authors to do:
At the end of the abstract, it is worthwhile to present your best-obtained results in terms of performance metrics as percentages or values.
In the introduction section or where appropriate, you may need to cite and add the following recent references regarding blockchain technology and vehicle detection, and intelligent automotive systems:
Razaque, A.; Bektemyssova, G.; Yoo, J.; Alotaibi, A.; Ali, M.; Amsaad, F.; Amanzholova, S.; Alshammari, M. Malicious Vehicle Detection Using Layer-Based Paradigm and the Internet of Things. Sensors 2023, 23, 6554. https://doi.org/10.3390/s23146554
Shakerian, A.; Eghmazi, A.; Goasdoué, J.; Landry, R.J. A Secure ZUPT-Aided Indoor Navigation System Using Blockchain in GNSS-Denied Environments. Sensors 2023, 23, 6393. https://doi.org/10.3390/s23146393
Alshraideh, M.; Mahafzah, B.; Al-Sharaeh, S.; Hawamdeh, Z. A Robotic Intelligent Wheelchair System Based on Obstacle Avoidance and Navigation Functions. Journal of Experimental & Theoretical Artificial Intelligence 2015, 27, 471–482. https://doi.org/10.1080/0952813X.2014.971441
In Section 4 and before Subsection 4.1, write one small overview paragraph about Section 4 and its subsections.
You may need to mention the machine hardware specifications (CPU/GPU, cache, RAM, etc.) and operating system, you did use in implementing your techniques, even if they don’t have a great impact on the performance of the implemented techniques.
The obtained results in the figures need to be explained and justified. That is, it is worthwhile to justify the results concerning the effect of the proposed technique from the algorithmic design point of view.
The quality of the English language is good. The authors may need to check the whole manuscript for grammar, spelling, and formatting issues in general.
Author Response
We thank the reviewer for their invaluable suggestions to improve our manuscript. We have revised it accordingly.

Reviewer 2 Report
This study discusses the development of a Block-chain based Collaborative Distributed Intrusion Detection System with a novel dynamic throttling strategy to detect and prevent attackers’ lateral movement among devices in vehicular social networks. Using this approach, it is shown that the proposed platform can be applied to the new vehicular social networks context with some modifications to deal with the large number of nodes. The work has been prepared well and contains a comprehensive background research and the details of the study were explained clearly. The work has a potential to be published here, however, there are some comments and concerns that should be addressed in the revised version of the manuscript. Here are my comments and suggestions:
1) The precision of the proposed approach is not clear to me. The authors should conduct a quantitative comparison or qualitative evaluation of the proposed mechanism.
2) The cost and expenses of the proposed approach should be discussed and compared.
Author Response

(The authors gave the same response as above.)

Reviewer 3 Report
The paper called Efficient Blockchain-based Intrusion Detection System with Novel Dynamic Throttling Strategy for 6G-enabled VSNs by Lampis Alevizos, Vinh Thong Ta and Max Hashem Eiza. The paper is very good; there are only few small improvements to make. There are some major aspects I would like to highlight. There are some things that could be added to the paper to broaden the scope of the paper along with the group of potential readers.
Very good research work, requires a few additions and corrections;
1) The title should be redrafted to reflect the nature of the work.
2) The abstract should be written in accordance with MDPI standards.
3) please adjust the publication to MDPI standards so that it includes:
Introduction,
Materials and Methods,
Results,
Discussion,
Conclusions,
4) In the abstract presented, the importance of the publication should be more concisely described including more extensive consideration of the methods, analyses and results of the research obtained.
5) It would be necessary to emphasize what is the main scope and purpose of the presented publication, it is difficult to find it in the text,
6) What this work contributes to science.
The presented conclusions may be of fundamental importance, therefore they should be presented in a better light and the author(s) should emphasize the original research contribution. I believe, that suggested amendments will significantly increase the relevance of the publication and will improve it. After applying all required changes, the paper is suitable for publication.
Author Response

(The authors gave the same response as above.)

Round 2
Reviewer 3 Report
Thank you for the changes you made
Now the publication is correct
Accept in present form